# Systemic Inflammatory Molecules Are Associated with Advanced Fibrosis in Patients from Brazil Infected with Hepatitis Delta Virus Genotype 3 (HDV-3)

**DOI:** 10.3390/microorganisms11051270

**Published:** 2023-05-12

**Authors:** Mauricio Souza Campos, Juan Miguel Villalobos-Salcedo, Deusilene Souza Vieira Dallacqua, Caio Lopes Borges Andrade, Roberto José Meyer Nascimento, Songeli Menezes Freire, Raymundo Paraná, Maria Isabel Schinoni

**Affiliations:** 1Instituto Ciências da Saúde, Universidade Federal da Bahia, Salvador 40231-300, Brazil; 2Programa de Pós-Graduação em Processos Interativos de Órgãos e Sistemas, Instituto Ciências da Saúde, Universidade Federal da Bahia, Salvador 40231-300, Brazil; 3Fundação Oswaldo Cruz (FIOCRUZ), Porto Velho 76812-245, Brazil; 4Programa de Pós-Graduação em Imunologia, Instituto Ciências da Saúde, Universidade Federal da Bahia, Salvador 40231-300, Brazil; 5Laboratório de Imunologia e Biologia Molecular, Instituto Ciências da Saúde, Universidade Federal da Bahia, Salvador 40231-300,Brazil; 6Hospital Universitario Professor Edgard Santos, Universidade Federal da Bahia, Salvador 40110-060, Brazil

**Keywords:** hepatitis D, systemic inflammatory molecules, fibrosis

## Abstract

Background and Aims: Hepatitis Delta virus (HDV) genotype 3 is responsible for outbreaks of fulminant hepatitis in Northeastern South America. This study investigates if systemic inflammatory molecules are differentially expressed in patients with advanced fibrosis chronically infected with Hepatitis Delta virusgenotype 3(HDV-3). Methods: Sixty-one patients from the north of Brazil coinfected with hepatitis B virus (HBV)/HDV-3 were analyzed. HDV quantification and genotyping were performed by semi-nested real-time polymerase chain reaction (RT-PCR) and restriction fragment length polymorphism (RFLP) methodologies. Ninety-two systemic inflammatory molecules (SIMs) were measured by Proximity Extension Assay (PEA) technology. The Shapiro–Wilk, Student’s *t*-test, Mann–Whitney tests, and logistic regression analysis were used when appropriate. Results: The median age was 41 years, and all patients were HBeAg negative. Advanced fibrosis or cirrhosis was diagnosed by histological staging in 17 patients, while 44 presented with minimal or no fibrosis. Advanced necroinflammatory activity correlated positively with serum levels of aspartate aminotransferase (AST) and alanine aminotransferase (ALT). Established non-invasive fibrosis scores (APRI, FIB-4, and AST/ALT ratio) revealed low sensitivities and positive predictive values (PPVs) with an AUROC maximum of 0.586. Among the 92 SIMs analyzed, MCP.4, CCL19, EN.RAGE, SCF, and IL18 showed a positive correlation with fibrosis stage. A combined score including CCL19 and MCP.4 revealed a sensitivity of 81% and an odds ratio of 2.202 for advanced fibrosis. Conclusions: Standard non-invasive fibrosis scores showed poor performance in HDV-3 infection. We here suggest that the determination of CCL19 and MCP.4 may be used to identify patients with advanced fibrosis. Moreover, this study gives novel insights into the immunopathogenesis of HDV-3 infection.

## 1. Introduction

The Hepatitis Delta virus (HDV) is a single ribonucleic acid (RNA) pathogen that requires B virus surface antigen (HbsAg) for assembly and viral transmission [1]. Studies have shown that Hepatitis B Virus and Hepatitis D Virus (HBV-HDV) coinfection can accelerate liver diseases and that HBV-HDV superinfection may result in acute fulminant hepatitis or chronic hepatitis, evolving in many cases to cirrhosis and hepatocellular carcinoma [1,2]. It is estimated that about 240 million people are HbsAg positive worldwide, and of these, 2–8% are coinfected with HDV, resulting in 10 to 20 million people suffering from delta hepatitis [3]. The prevalence of hepatitis delta is significantly different among regions of the world [3]. At least eight different types of Delta virus were identified based on the diversity of their genomes and their specific geographic distribution [4,5]. HDV-1 is distributed in Europe, the Middle East, America, and North Africa [4]. HDV-2 to HDV-8 occur in specific regions [4]. HDV-3, described as the most divergent and aggressive genotype, is exclusively found in South America, specifically in the Western Amazon basin, including Brazil, Peru, Ecuador, Venezuela, and Colombia, and it is responsible for outbreaks of severe and common fulminant hepatitis in this area [2,6,7]. In Brazil, this area corresponds to the states of Acre, Amazonas, Rondônia, and Roraima, with a significant prevalence in the indigenous population [8].

Viral infections induce the body to mount a complex immune response, involving components of the innate and adaptive immune systems [9]. Immune mediation has been shown to be generally implicated in the liver damage associated with HDV. The pathogenesis of HDV mainly involves the activation of HDV-specific T lymphocytes, natural killer (NK) cells, cytokine responses, and NF-kB signaling [10]. HDV can also cause direct hepatocyte injury or hepatic tissue inflammation. The most likely cause of these effects is the immune response of the host against the pathogen. In general, chronic viral infections are characterized by the presence of antiviral T cells, but they are often unable to control viral replication and the progression of infection. Schirdewahn et al. (2017) demonstrated in a study with 92 patients, 49 of whom were coinfected with HBV-HDV, that functional recovery of these HDV-specific T cells may be possible if there is targeting of signals mediated by T cell receptors (signal 1), a costimulatory signal (signal 2), and a cytokine signal (signal 3) that are increasingly recognized as being essential in the control of viral infection [10]. Until now, there have been no studies demonstrating the association between these soluble inflammatory molecules and liver damage caused by HDV, including HDV-3 [11].

Markers of fibrosis stages in patients with chronic hepatitis D genotype 1 have been investigated in order to standardize a score using non-invasive methods. To date, all validated scales (FIB-4, APRI, and Relation AST/ALT) to measure the grade of fibrosis in patients with advanced liver disease caused by other etiologic agents have proved unsatisfactory for assessing fibrosis in patients with chronic hepatitis D. Lutterkort et al. (2017), in a study of 100 HDV-RNA-positive genotype 1 patients, developed a new model to evaluate fibrosis in patients with Delta hepatitis based on levels of cholinesterase, albumin, and gamma-glutamyl transferase as well as the age of the patient. The Delta Fibrosis Score (DFS) showed a sensitivity of 85% and a positive predictive value (PPV) of 93%, with an AUROC of 0.87 [12]. Similar findings were obtained in a study by Takyar (2017), which aimed to evaluate the safety of percutaneous liver biopsy performed with a Klatskin needle [13]. However, the new scale to evaluate the degree of fibrosis in the Delta virus infection still requires the establishment of new parameters, and it was not tested and validated for patients infected with HDV-3.

Thus, in the present study, our objective was to investigate the association between systemic inflammatory molecules and advanced fibrosis in patients chronically infected with HDV-3 and to offer a new non-invasive model to evaluate the stage of fibrosis in these patients.

## 2. Patients and Methods

### 2.1. Patient Cohort

In total, 61 patients coinfected with HBV and Delta virus genotype 3 diagnosed between 2002 and 2016 were included in this study. All patients were older than 18 years old, and we included patients of both sexes. Patients coinfected with Hepatitis C Virus (HCV), Human T-Lymphotropic Virus (HTLV) I and II, and/or Human Immunodeficiency Virus (HIV) I and II, or with autoimmune or rheumatologic diseases that could affect the serum profile of cytokines, were excluded. All participants were informed of the objectives of the project, and after explanation, they signed a Free and Informed Consent Term (TCLE) prepared especially for this purpose. Clinical and epidemiological data were collected from patients’ records when available. This study was conducted in conformity with the principles of the Declaration of Helsinki. Written informed consent was obtained in all cases. The study was a priori approved by the local ethics committee of the Institute of Health Sciences of the Federal University of Bahia, Brazil.

### 2.2. Samples and Laboratory Assays

About 10 mL of whole blood were collected from each participant in a tube without an anticoagulant. After centrifugation at 3500 rpm for 10 min, aliquots of serum, approximately 1 mL, were stored at −20 °C for serological tests and −70 °C for molecular tests.

Diagnostic tests for HBV and HDV infections were performed using enzyme-linked immunosorbent assays (ELISA) with commercial kits. All serums were examined for HBsAg (OrganonTeknika, Amsterdam, The Netherlands), antibody against viral core antigen (anti-HBc) (OrganonTeknika, Amsterdam, The Netherlands), antibody against hepatitis B surface antigen (anti-HBs) (Abbott, Chicago, IL, USA), specific antibody against the hepatitis B virus (anti-HBe), immunoglobulin M antibody against viral core antigen of the hepatitis B virus (anti-HBcIgM), and antibody against the hepatitis D virus (anti-HDV) (OrganonTeknika, Amsterdam, The Netherlands).

### 2.3. Viral Quantification and Genotyping

The deoxyribonucleic acid of the hepatitis B virus (HBV DNA) was quantitatively screened using commercial kits from the Roche laboratory (CobasAmplicor HBV Monitor Test Switzerland), which relies on the amplification and hybridization of viral DNA. The detection and quantification of HDV RNA wereperformed through the technique of semi-nested RT-PCR, according to the methodology described by Casey et al. (1996). Genotyping of HDV RNA was performed using Restriction Fragment Length polymorphism (RFLP) methodology. The amplified product obtained was digested with the restriction enzyme SmaI (Invitrogen Life Technologies) in the following reaction: fivemicroliters of the PCR product were incubated for 2 h at 37 °C with the 5U restriction enzyme. This technique allows the specific detection of PCR products for genotypes I, II, and III [14].

### 2.4. Soluble Inflammatory Mediators (SIMs) Measurements

This panel has been used to determine whether diabetic patients with abdominal aortic aneurysms have a distinct plasma inflammatory profile compared to non-diabetic patients [15]. Ninety-two SIMs were measured by Proximity Extension Assay (PEA) technology (Proseek Multiplex Inflammation I assay). This technology allows the simultaneous measurement of 92 inflammatory-related human proteins based on Proximity Extension Assay Technology [15]. The Proseek Multiplex Inflammation I kit has been shown to have high reproducibility and repeatability, with mean intra- and inter-assay coefficients of variation of 7% and 18%, respectively. The data analysis followed a normalization procedure. For each data point, delta Cq values were obtained by subtracting the value of the extension control. Normalization between runs was performed by subtracting the interplate control for each assay. The values were then set relative to a correction factor determined by Olink to obtain normalized protein expression (NPX). The data were finally expressed as 2NPX. This provides a relative protein expression, where a highprotein value corresponds to a highprotein concentration.

### 2.5. Statistical Analyses

We used SPSS Statistics 17 (Windows & Mac) statistical software. For the descriptive analysis, the quantitative variables were represented by their means and standard deviations when their distributions were normal and by medians and interquartiles when they were not normal. The definition of normality was made through graphical analysis and the Shapiro–Wilk test. Categorical variables were represented by frequencies and percentages. Bivariate comparisons between groups were made using the Student’s *t*-test for normal distribution variables and the Mann–Whitney test for non-normal distribution variables. Logistic regression analysis was implemented. Independent variables for the final model were selected via backwise stepwise.

## 3. Results

### 3.1. Clinical Characteristics of Study Subjects

In this study, 61 patients coinfected with HBV/HDV-3 were evaluated.Seventeen patients with advanced stages of fibrosis or cirrhosis and forty-four with minimal or no fibrosis. As all patients were from an endemic region, there was no association with risk factors such as the use of intravenous drugs, blood transfusions, or tattoos. No patient was working as a healthcare professional. The median age of the participants was 41 years (18–59 years). Hepatitis D virus was not positively associated with age (*p* = 0.1056), and 39 patients were male (63.9%). Baseline characteristics are shown in Table 1. No correlation was found between alcoholic beverage use and the stage of necroinflammatory activity. Advanced necroinflammatory activity was positively correlated with serum levels of AST and ALT (*p* = 0.020 and *p* = 0.024, respectively). Serum levels of gamma-glutamyl transferase (GGT), total and indirect bilirubin, as well as platelets, showed no association with the fibrosis stage of the patients studied. Serological tests were performed pre-treatment, and all patients presented positive HBsAg and negative HBV surface antibodies (anti-HBs). All patients presented negative HBeAg and positive anti-HBe.

### 3.2. Soluble Inflammatory Mediators in Non-Cirrhotic vs. Cirrhotic Patients

We performed a multi-analytic profile of 92 soluble inflammatory mediators (SIMs): chemokines, adhesion molecules, and growth factors in the blood serumof all samples by PEA technology (Proseek Multiplex Inflammation I assay). Of these, 74 were effectively analyzed. The expression pattern of all analyzed SIMs in serum samples presented varied results between patients with advanced fibrosis or cirrhosis and patients with minimal or no fibrosis. Among the 74 SIMs analyzed, five showed a positive correlation with fibrosis stage: MCP.4 (*p* = 0.032), CCL19 (*p* = 0.024), EN.RAGE (*p* = 0.014), SCF (*p* = 0.01), and IL18 (*p* = 0.054). TNF beta (*p* = 0.098) showed a tendency to correlate with the parameter described previously (Table 2). All other SIMs did not show a significant correlation with the development of fibrosis in HDV-3 patients. Although most systemic inflammatory molecules did not show a positive correlation with the stage of liver disease in the patients studied (*p* > 0.05), it was possible to observe higher serum levels of these inflammatory markers in patients with advanced fibrosis or cirrhosis than in patients with minimal or non-fibrosis, as can be seen in Figure 1. When comparing the patients classified in the METAVIR score as FO, FI, F2, F3, and F4, it can be seen that patients with a more advanced stage of fibrosis or cirrhosis present an altered inflammatory cytokine and chemokine milieu. Some patients classified as F0, F1, and F2 on the METAVIR scale presented higher serum levels of inflammatory cytokines and chemokines than expected for this hepatic necroinflammatory stage (Figure 1). Likewise, only two patients classified as F3 as a result of a biopsy showed levels of SIMs that were not expected. Figure 2 shows that soluble inflammatory mediators that correlate with the degree of fibrosis in the patients studied are functionally responsible for the recruitment and activation of cells of the innate immune system (macrophages and NK cells), fibrogenesis, and direct mediators of the inflammatory process of tissues, inducing or inhibiting the production of pro-inflammatory cytokines. CCL19 (*p* = 0.024) and IL18 (*p* = 0.054) were higher in patients with advanced fibrosis grade, whereas EN.RAGE (*p* = 0.014), SCF (*p* = 0.01), and MCP.4 (*p* = 0.032) presented higher levels among patients with low or no degree of fibrosis (Table 2).

### 3.3. Clinical Markers of Fibrosis in HDV-Infected Patients

We evaluated in HDV-3 patients the performance of existing non-invasive systems for evaluating the fibrosis stage of other etiologies using cut-off points described in the literature. In general, scores developed to evaluate the stage of fibrosis in patients with viral hepatitis (APRI, FIB-4, and AST/ALT) revealed low sensitivities and PPVs with an AUROC maximum of 0.586 for the APRI and FIB-4 and 0.503 for the AST/ALT. Although the results showed a relatively high specificity in the three analyzed scores, none revealed a predictive value (PV) of more than 50%.

### 3.4. ROC Curve SIMs Considerations

A logistic regression analysis was performed to compare hepatic biopsy results, considered the gold standard for the definition of degree of fibrosis in patients with chronic hepatic disease, with the SIMs quantified between the groups of patients with advanced-stage fibrosis and patients without fibrosis or in an early stage of disease. The independent variables for the final model were selected via backwise stepwise. It can be seen in Figure 3 that the cytokine CCL19 and the chemokine MCP.4, when analyzed together, presented an AUROC with a sensitivity of fibrosis stage differentiation of 80.9% when compared to the liver biopsy performed in the patients. Logistic regression also showed that the association between CCL19 (*p* = 0.004) and MCP.4 (*p* = 0.032) presented an odds ratio (OR) of 2.202.

## 4. Discussion

Chronic delta hepatitis is described as the most serious form of viral hepatitis, and it often evolves into cirrhosis and hepatocellular carcinoma (HCC) [15]. To date, few studies have been conducted with antiviral drugs tested exclusively in patients infected with HDV-1. HDV-3 is known to have the greatest genotype divergence, and it is found only in South America [16]. It is described as the most aggressive among the eight genotypes described in HDV, and it has a high prevalence in the Brazilian Amazon region [17]. Some recent studies focusing on the pathophysiology of HDV infection have demonstrated that the host’s immune response against the virus is directly linked to liver tissue damage as well as to the clinical course of the disease, but unfortunately, to date, no study has evaluated the importance of soluble inflammatory molecules in the clinical outcome of patients infected by HDV [9,10,12,17]. This study provides the first comprehensive analysis of the association between systemic inflammatory molecules and advanced fibrosis in patients with hepatitis D infected with HDV-3.

The study population consisted of 61 individuals, with a general average age of 41 years (39 years among patients without fibrosis and 44 years among fibrosis patients), suggesting a low life expectancy at birth. Males presented a higher rate of infection, on average twice as high as women, but when the degree of fibrosis between the two groups was compared, the proportion of advanced fibrosis in men and women was very close (30.6% and 22.8%, respectively). Other studies performed among native Amazonian populations showed the same pattern of distribution described [18]. Patients in more advanced hepatic disease stages (advanced fibrosis) had higher serum AST and ALT levels than patients who did not present fibrosis (*p* = 0.024 and *p* = 0.020, respectively). These findings are in agreement with previous studies that have evaluated the same pattern in different populations [8,11]. Interestingly, serum GGT levels did not present statistically significant differences, contrary to previous studies [8,11,19].

All patients analyzed were HBeAg negative. These data diverge from a previous study conducted by Braga et al. (2001), who found a much higher positive HBeAg ratio among HBV/HDV coinfected patients. This result, however, confirms that HDV reproduction plays a crucial role in the modulation of HBV replication [20,21,22,23].

### 4.1. SIMs Associated Fibrosis

In the present study, of the seventy-four systemic inflammatory molecules analyzed, five were statistically significantly associated with the stage of liver fibrosis: MCP.4 (*p* = 0.032), CCL19 (*p* = 0.024), EN.RAGE (*p* = 0.014), SCF (*p* = 0.01), and IL18 (*p* = 0.054). To date, there have been no studies relating SIMs to the degree of fibrosis in Delta hepatitis; thus, these findings are novel.

A study of asthmatic patients divided into two groups: controlled asthma and exacerbated asthma, showed a positive correlation between elevated levels of MCP.4 and exacerbation of the disease, revealing the role of this chemokine in the systemic inflammatory process [24]. In this study, MCP.4 levels in patients with advanced fibrosis were lower than in patients without fibrosis or at an early stage of fibrosis. Paradoxically, while in asthmatic patients, high levels of MCP.4 are implicated in exacerbation of the disease, in individuals with HDV-3, this chemokine appears to have a protective effect, decreasing inflammatory activity.

CCL19 is a cytokine directly involved in the attraction of cells of the innate immune system, mainly dendritic cells, to the site of inflammation, the feedback process, and contributing to the exacerbation of inflammatory activity [25]. In vitro studies have demonstrated that CCL19 induces the activation and proliferation of immune cells of the innate system by increasing the levels of pro-inflammatory cytokines. The results of this study showed that CCL19 levels in cirrhotic patients are higher than in patients in the early stages of fibrosis. The attraction of innate immune system cells to the liver during infection by HDV-3 appears to be a key factor in the evolution of hepatic tissue damage.

Although the results of the MCP.4 and CCL19 levels found in this study seem contradictory since both act on the recruitment of cells from the innate immune system to the site of infection, they suggest that systemic inflammatory molecules act together, modulating the effects of each other and acting differently in various diseases.

Hasegawa et al., (2004), studying the regulation of EN.RAGE expression in human macrophages, demonstrated that inflammatory cytokines, like IL6, increase EN.RAGE expression. EN.RAGE in resident macrophages contributes to the maintenance of the inflammatory process [26]. In this study, although IL-6 levels were higher in patients with advanced fibrosis, the levels of EN.RAGEwere higher in the group of patients with early-stage or non-fibrosis. These results suggest that in HDV-3 infection, the activity of systemic inflammatory molecules is influenced by other unknown factors and demonstrate the importance of new studies to clarify the pathophysiology of the infection.

SCF (stem cell factor) is a pleiotropic cytokine exerting its role at different stages of bone marrow development and affecting eosinophil activation, mast cells, and basophil chemotaxis and survival. Previous studies showed increased production of SCF in different allergic diseases like asthma, allergic rhinitis, and atopic dermatitis [27]. Expression of SCF and its receptor c-kit mRNA in the airway epithelium of patients with asthma was increased in comparison to healthy control airways. In our study, SCF levels were higher in patients with early fibrosis or without fibrosis than in patients with advanced fibrosis. These divergent results in inflammatory diseases seem to confirm the pleiotropic nature of this cytokine.

Sharma et al., (2009) conducted a study to investigate the possible role of IL18 in the pathogenesis and persistence of HCV. IL18 levels were measured in serum from 50 patients at various stages of HCV infection (resolved, chronic, and cirrhotic) and compared to those from normal controls. IL18 gene expression was studied in peripheral blood mononuclear cells (PBMC) in each group and in hepatic biopsy tissue from patients with chronic hepatitis C. Mean serum IL18 levels were markedly elevated in patients with chronic hepatitis and cirrhosis and were reduced in patients with a resolved HCV infection. Serum IL18 concentrations were related to the severity of Child-Pugh liver disease in cirrhotic patients. There was also a strong positive correlation between IL18 levels, histological activity, and necrosis scores. In our study, serum IL18 levels were also higher in patients with advanced fibrosis, suggesting a relationship between this cytokine and the inflammatory activity of the liver [28].

### 4.2. Established Fibrosis Scores

Chronic hepatitis Delta continues to be a disappointment in medical practice, as it is the most serious form of hepatitis among humans’ hepatitis viruses, often evolving to cirrhosis and hepatocellular carcinoma [29]. Available treatments have been shown to be poorly effective in controlling disease progression. In recent years, some clinical trials using new drugs have been conducted with seemingly promising results. While new effective therapies do not emerge, monitoring of the fibrosis stage in patients with chronic hepatitis D is necessary as a way of identifying patients with advanced fibrosis or cirrhosis who require immediate medical intervention [13]. The gold standard in the staging of fibrosis and necroinflammatory activity continues with liver biopsy; however, because it is an invasive method with relevant risks, it is not used for serial follow-up of the patients [13]. An alternative to this would be the development of non-invasive fibrosis assessment scores. Classically used methods, developed for other liver diseases, failed to assess the patients in this study, revealing low sensitivity and PPVs with a maximum AUROC of 0.586. These results demonstrate that existing non-invasive fibrosis stage evaluation methods are not effective for the evaluation of chronically infected patients with HDV. These results are consistent with other studies investigating the performance of these scores in various liver diseases and show the need for studies to identify possible biomarkers of hepatic fibrosis stage in patients with chronic hepatitis D [12,13,28,29].

### 4.3. Alternative Fibrosis Scores Based on SIMs

All the above-mentioned studies corroborate the findings in this cohort and suggest a correlation between the level of cytokines/chemokines and the inflammatory activity in several disease models. Our study also suggests that some cytokines are implicated in the pathophysiology of hepatic fibrosis in patients with chronic HDV-3 and can be used as biological markers for the identification of advanced fibrosis stages. The combined use of serum levels of CCL19 and MCP.4 has so far proved to be the most sensitive non-invasive method for the detection of advanced fibrosis in patients chronically infected with HDV-3.

### 4.4. Influencing Factors—DNA HBV and RNA HDV

It is important to emphasize that some studies have demonstrated the influence of HBV and HDV viral loads in coinfected patients on serum levels of cytokines and, consequentlyon the clinical evolution of the disease. In this study, the viral loads of HBV and HDV in patients were not analyzed.

### 4.5. Limitations

The main limitations of our work were that only one pathologist evaluated the hepatic tissue samples; therefore, it is not possible to compare different analyses of the same material; furthermore, HDV-3 is not fully defined, and there are few studies conducted with it [1,2]. Another relevant point is the large number of SIMs analyzed in this series. Some SIMs could not be evaluated for technical reasons (the cut-off points for reading were not reached). None of the parameters analyzed in this study were validated by previous studies with cohorts of patients with HDV-3, so we cannot compare the results of this study with others. The fibronectin discriminant score (FDS) based on fibronectin predicted liver fibrosis with a high degree of accuracy in patients infected with other hepatotropic viruses, potentially decreasing the number of liver biopsy required, but, unfortunately, there are also no studies of the accuracy of this score in the evaluation of the degree of fibrosis in patients infected with HDV-3. Thus, future studies will need to investigate in more detail the histological features of hepatitis delta genotype 3 in correlation with clinical and virological parameters.

## 5. Conclusions

To date, there are no studies relating systemic inflammatory molecules to the degree of fibrosis in HDV-3 Hepatitis Delta patients. Standard non-invasive fibrosis scores showed poor performance in HDV-3 infections. This makes the follow-up of these patients a challenge for clinicians. We suggest here that the determination of CCL19 and MCP.4 can be used to identify patients with advanced fibrosis. In addition, this study provides new insights into the immunopathogenesis of HDV-3 infection.

## Figures and Tables

**Figure 1 microorganisms-11-01270-f001:**
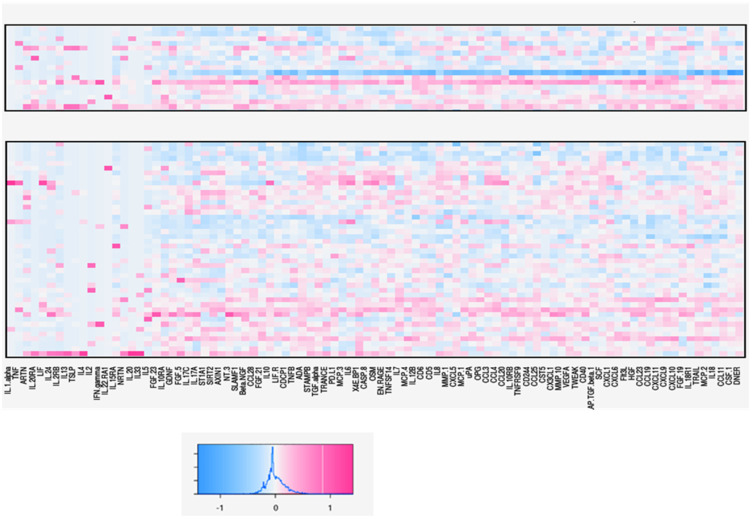
SIMs Heat Map. Proximity Extension Assay (PEA) technology (Proseek Multiplex Inflammation I Assay). The upper region contains Advanced Fibrosis (n = 17), and the lower region contains Non-Cirrhotic (n = 44) results. Each line represents a patient, and each column represents a SIM.

**Figure 2 microorganisms-11-01270-f002:**
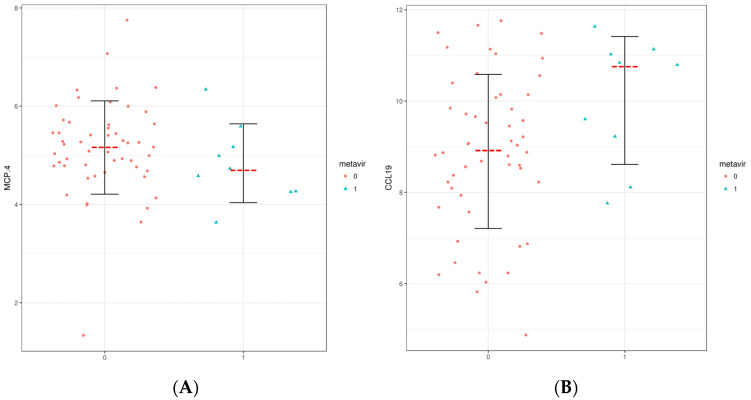
Statistically significantSIMs. (**A**) MCP.4 (*p* = 0.032). (**B**) CCL19 (*p* = 0.024). (**C**) EN.RAGE (*p* = 0.014). (**D**) SCF (*p* = 0.01). (**E**) IL18 (*p* = 0.054).

**Figure 3 microorganisms-11-01270-f003:**
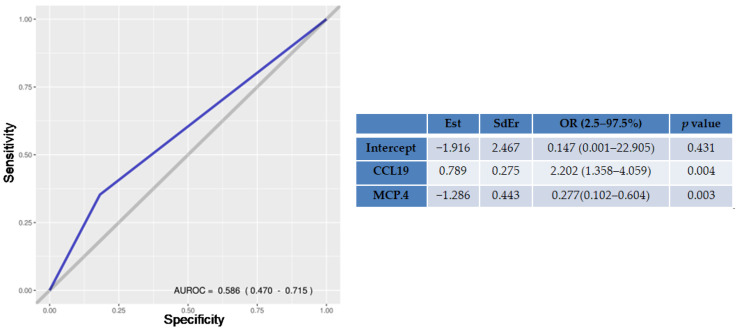
SIMs AUROC Curve. Est, estimate; SdEr, standard error; OR, odds ratio.

**Table 1 microorganisms-11-01270-t001:** Clinical characteristics of study subjects.

Characteristic	Total (n = 61)	Non-Cirrhotic (n = 44)	Advanced Fibrosis (n = 17)	*p*-Value
Gender				
Female	22 (36.1%)	17 (38.6%)	5 (29.4%)	
Male	39 (63.9%)	27 (61.4%)	12 (70.6%)	
Age Median=(interval=)	41 (18–59)	39 (18–57)	44 (18–59)	0.1056
Use of alcohol				
No	45 (73.8%)	32 (72.7%)	13 (76.5%)	
Yes	16 (26.2%)	12 (27.3%)	4 (23.5%)	
ALT Level (U/L) Median=(interval=)	19.0 (11.0–29.0)	17.0 (10.0–26.2)	28.0 (17.0–36.0)	0.024
AST Level (U/L) Median=(interval=)	20.0 (12.0–26.0)	17.0 (10.8–23.0)	28.0 (13.0–32.0)	0.020
GGT Level (U/L) Median=(interval=)	21.0 (13.0–31.0)	20.5 (14.8–30.0)	26.0 (9.0–35.0)	0.929
TB* (mmol/L) Median=(interval=)	13.0 (10.0–21.0)	13.0 (10.5–19.0)	21.0 (10.0–24.0)	0.083
AFP** (ng/mL) Median=(interval=)	25.0 (13.0–39.0)	30.5 (16.8–39.2)	21.0 (5.0–25.0)	0.012
Platelets (109/L) Median=(interval=)	21.0 (9.0–34.0)	22.5 (9.8–34.2)	18.0 (9.0–34.0)	0.778
ALB Median=(interval=)	17.0 (10.0–23.0)	17.0 (12.8–24.2)	11.0 (8.0–18.0)	0.085
PT Median=(interval=)	12.0 (8.0–20.0)	14.0 (8.0–20.5)	9.0 (7.0–16.0)	0.143
HBV DNA Median=(interval=)	348,786.257 (0.0–18,700,000)	101,425.0 (0.0–2,295,572)	113 (0.0–745)	0.3468

ALT, alanine aminotransferase; AST, aspartate aminotransferase; GGT, gamma glutamyl transferase; TB*, total bilirubin; AFP**, alpha-fetoprotein; ALB, albumin; PT, prothrombin time.

**Table 2 microorganisms-11-01270-t002:** Statistically significant SIMs. SIMs in Non-Cirrhotic vs. Advanced Fibrosis.

SIMs	Total (n = 61)	Non-Cirrhotic (n = 44)	Advanced Fibrosis (n = 17)	*p*-Value
SCF	9.4 (8.8–9.8)	9.6 (9.1–9.9)	8.4 (8.0–9.5)	0.01
EN.RAGE	5.5 (4.2–5.9)	5.8 (5.2–5.9)	4.8 (4.2–5.1)	0.014
CCL19	9.1 (8.1–10.4)	8.8 (7.9–9.8)	10.1 (9.0–11.0)	0.024
MCP.4	5.1 (4.7–5.6)	5.3 (4.8–5.7)	4.7 (4.3–5.2)	0.032
IL18	8.5 (8.0–9.0)	8.4 (7.9–8.9)	8.8 (8.5–9.3)	0.054

## Data Availability

The data presented in this study are available in the presente article.

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
