# Peer review of "Systemic Inflammatory Molecules Are Associated with Advanced Fibrosis in Patients from Brazil Infected with Hepatitis Delta Virus Genotype 3 (HDV-3)"

_microorganisms, 2023, doi:10.3390/microorganisms11051270_

Round 1

Reviewer 1 Report

SIM are novel in Liver diseases  and fibrosis and even more in hdv hepatitis where  it is especially relevant since usual non invasive markers failed in HDV .This  should be further illustrated and commented in the discussion both with bLOOD TESTS  BUT ALSO WITH FIBROSCAN WHICH IS A UNIQUE AND  MAJOR PITFALL ! this does  open the way to relevance of  SIM  

The EASE and availability/COST  of SIM must be discussed especially in the context of RLC 

DISCUSSION is  too long and  diverse when it  should focus on pathogenic mechanisms and most convincing relevance of SIM 

Author Response

1) SIM are novel in Liver diseases  and fibrosis and even more in hdv hepatitis where  it is especially relevant since usual non invasive markers failed in HDV .This  should be further illustrated and commented in the discussion both with bLOOD TESTS  BUT ALSO WITH FIBROSCAN WHICH IS A UNIQUE AND  MAJOR PITFALL ! this does  open the way to relevance of  SIM.

Answer 1) ALREADY DISCUSSED IN THE PAPER

     "The gold standard in staging of fibrosis and necroinflammatory activity continues by liver biopsy, however, because it is an invasive method with relevant risks, it is not used for serial follow-up of the patients.13An alternative to this would be the development of non-invasive fibrosis assessment scores. Classically used methods, developed for other liver diseases, failed to assess the patients in this study, revealing low sensitivity and PPVs with a maximum AUROC of 0.586. These results demonstrate that existing non-invasive fibrosis stage evaluation methods are not effective for the evaluation of chronically infected patients with HDV. These results are consistent with other studies investigating the performance of these scores in various liver diseases and show the need for studies to identify possible biomarkers of hepatic fibrosis stage in patient with chronic hepatitis D.12,13,28,29"

2) The EASE and availability/COST  of SIM must be discussed especially in the context of RLC

3) DISCUSSION is  too long and  diverse when it  should focus on pathogenic mechanisms and most convincing relevance of SIM 

Answer 2 AND 3) ALREADY DISCUSSED IN THE PAPER

"All the above mentioned studies corroborate with the findings in this cohort and suggest a correlation between the level of cytokines/chemokines and the inflammatory activity in several disease models. Our study also suggests that some cytokines are implicated in the pathophysiology of hepatic fibrosis in patients with chronic HDV-3 and can be used as biological markers for the identification of advanced fibrosis stage.  The combined use of serum levels of CCL19 and MCP.4 has so far proved to be the most sensitive non-invasive method for the detection of advanced fibrosis in patients chronically infected with HDV-3."

Reviewer 2 Report

In this study, authors investigated the expression of systemic inflammatory molecules in HDV-3 patients. They found that CCL19 and MCP.4 may be used to identify patients with advanced fibrosis. Their discussion about the findings was comprehensive. But some issues still need to be addressed.

1. Abbreviations should be used in full names the first time they appear, even in the Abstract.

2. The language quality needs to be improved. Proofreading is suggested.

3. Line 50: Please clarify the “specific regions”.

4. Line 61: Please cite references to support “HDV does not cause direct hepatocyte injury or hepatic tissue inflammation”. Some studies have shown that HDV can cause direct cytopathic damage during acute infection (PMID: 33924806).

5. Line 137: Please specify the software used for statistical analyses.

6. Please provide the sequence access number of HDV-3 isolated in this study.

7. Why significant digits in Table 1 were different?

8. The quality of all figures should be improved.

9. The figure legend of Figure 3 should be detailed.

Author Response

1) Abbreviations should be used in full names the first time they appear, even in the Abstract.

Answer 1) Adjusted.

2) The language quality needs to be improved. Proofreading is suggested.

Answer 2) FULL REVISED TEXT.

3) Line 50: Please clarify the “specific regions”.

Answer 3) SPECIFIED IN THE TEXT.

4) Line 61: Please cite references to support “HDV does not cause direct hepatocyte injury or hepatic tissue inflammation”. Some studies have shown that HDV can cause direct cytopathic damage during acute infection (PMID: 33924806).

Answer 4) CORRECTED TEXT STATEMENT: HDV can also cause direct hepatocyte injury or hepatic tissue inflammation.

5) Line 137: Please specify the software used for statistical analyses.

Answer 5) IN TEXT: "We used Spss statistic software. For the descriptive analysis, the quantitative variables were represented by their means and standard deviations when their distributions were normal and by medians and interquartiles when not normal. The definition of normality was made through graphical analysis and Shapiro-Wilk test. Categorical variables were represented by frequencies and percentages. Bivariate comparisons between groups were made using Student's t test for normal distribution variables and the Mann-Whitney test for non-normal distribution variables.Logistic regression analysis was implemented. Independent variables for the final model were selected via backwise stepwise".

6) Please provide the sequence access number of HDV-3 isolated in this study.

Answer 6) WE DON'T HAVE

7) Why significant digits in Table 1 were different?

Answer 7) ALREADY CORRECTED IN THE TEXT

8) The quality of all figures should be improved.

Answer 8) THE IMAGES WERE WORKED ON AND THE ORIGINALS IMPROVED FOR THE ARTICLE

9) The figure legend of Figure 3 should be detailed.

Answer 9) THE LEGEND HAS THE NECESSARY INFORMATION FOR INTERPRETATION OF THE FIGURE

Reviewer 3 Report

I read with very interest the article entitled "Systemic inflammatory molecules are associated with advanced 2 fibrosis in HDV-3 Hepatitis Delta". 

The paper is well written and the topic is of interest given the recent development of drugs active against HDV. 

I have a few minor comments: 

-Paragraph 3.3. Clinical markers of fibrosis in HDV infected patients: please add some information in the method section regarding the assessment of fibrosis. Check  abbreviation for predictive value (PV and PPV in text). Did authors compare  noninvasive markers with liver biopsy? please specify in text. What about delta fibrosis score? you mentionned it in the introduction, did you evaluate PPV for delta fibrosis score with regard to liver biopsy?

Author Response

1) Paragraph 3.3. Clinical markers of fibrosis in HDV infected patients: please add some information in the method section regarding the assessment of fibrosis.

Answer 1) ALREADY IN THE TEXT: "Markers of fibrosis stages in patients with chronic hepatitis D genotype 1 have been investigated in order to standardize a score using non-invasive methods. To date, all validated scales (FIB-4, APRI, Relation AST / ALT) to measure grade of fibrosis in patients with advanced liver disease caused by other etiologic agents have proved unsatisfactory for assessing fibrosis in patients with chronic hepatitis D."

2) Check  abbreviation for predictive value (PV and PPV in text).

Answer 2) ALREADY CORRECTED IN THE TEXT

3) Did authors compare  noninvasive markers with liver biopsy? please specify in text.

Answer 3) NO. WE ONLY COMPARE THE SERUM LEVEL OF SIMS WITH LIVER BIOPSY. IT WAS NOT THE OBJECTIVE OF THE WORK TO DEVELOP THE DISCUSSION ABOUT OTHER NON-INVASIVE MARKERS, SINCE THE LITERATURE ALREADY ADMITS THAT THEY HAVE LOW SENSITIVITY IN THE EVALUATION OF PATIENTS WITH DELTA HEPATITIS.

4) What about delta fibrosis score? you mentionned it in the introduction, did you evaluate PPV for delta fibrosis score with regard to liver biopsy?

Answer 4) WE DO NOT EVALUATE THE VPP FOR THE DELTA SCORE.

Round 2

Reviewer 2 Report

The authors have revised the manuscript accordingly. The quality of this paper has improved a lot. It is suggested that the quality of figures should be further improved before publication can be considered.

Author Response

Thank you for your review, we changed all figures for it's original files from each software.